# Prediction of Fault Fix Time Transition in Large-Scale Open Source Project Data

**Hironobu Sone** [1,*] ⬤, **Yoshinobu Tamura** [2,†] ⬤ and **Shigeru Yamada** [3,†]

[1]  Graduate School of Integrative Science and Engineering, Tokyo City University, Setagaya, Tokyo 158-8557, Japan
[2]  Department of Intelligent Systems, Tokyo City University, Setagaya, Tokyo 158-8557, Japan
[3]  Graduate School of Engineering, Tottori University, Tottori, Tottori 680-8552, Japan
*  Correspondence: g1881827@tcu.ac.jp; Tel.: +81-3-5707-0104
†  These authors contributed equally to this work.

**Abstract:** Open source software (OSS) programs are adopted as embedded systems regarding their server usage, due to their quick delivery, cost reduction, and standardization of systems. Many OSS programs are developed using the peculiar style known as the bazaar method, in which faults are detected and fixed by developers around the world, and the result is then reflected in the next release. Furthermore, the fix time of faults tends to be shorter as the development of the OSS progresses. However, several large-scale open source projects encounter the problem that fault fixing takes much time because the fault corrector cannot handle many fault reports. Therefore, OSS users and project managers need to know the stability degree of open source projects by determining the fault fix time. In this paper, we predict the transition of the fix time in large-scale open source projects. To make the prediction, we use the software reliability growth model based on the Wiener process considering that the fault fix time in open source projects changes depending on various factors such as the fault reporting time and the assignees to fix the faults. In addition, we discuss the assumption that fault fix time data depend on the prediction of the transition in fault fixing time.

**Keywords:** reliability; open source software; transition of fault fixing time; stochastic differential equation; open source project

## 1. Introduction

The source code of open source software (OSS) is freely available for use, reuse, fixing, and re-distribution by the users. Many OSS programs are known for their high performance and reliability, even though they are free of charge. Furthermore, many IT companies often develop OSS for commercial use. In particular, OSS programs are developed using the bazaar method [1], in which the source code is disseminated to the public through the Internet. Then, OSS programs are promoted by an unspecified number of users and developers. The bug-tracking system is also one of the systems used to develop OSS. Much fault information such as fix status, the details, and fix priorities are registered through the bug-tracking system. Although OSS programs have been actively developed and used in recent years, there are problems with promoting open source projects. Massive open source projects have a large number of fault reports: there are over 100 faults reported per day in massive open source projects [2]. Due to this, it becomes difficult to fix the faults quickly [3,4].

From the perspective of the fixing speed of OSS, OSS users usually use the OSS under the conditions of stable development. There are many studies in terms of the stability degree assessment of open source projects [5–12]. There are many studies that predict the development effort in open source projects [5–8] and studies that predict the fix time of each reported fault [9–11]. Furthermore,

previous research showed that large open source projects tend to have a shorter fix time as the project progresses [12].

Generally, as the development of an OSS progresses, the number of unfixed faults and faults with a long fixing time decreases. Then, the OSS becomes stable. Several previous studies predicted the fault fixing time. However, these research works could not predict the software's stability, because the previous research did not consider the fixing time for all faults in software development.

The main contributions of the proposed method are as follows:

- By using the proposed method, the OSS manager can assess the stability of the OSS project considering the external factors of open source projects.
- Considering the Wiener process, the OSS manager can treat many noisy cases during the fault fixing time.
- The proposed method will lead to the assessment of the stability of OSS systems considering the convergence of the fault fixing time in various open source projects, because the proposed method can rapidly assess the stability and reliability of future projects by using fault fixing time data.

## 2. Methodology

In this paper, we predict that the time required for fault correction in OSS development decreases and converges with the transition of the fix time. By predicting the transition of the fix time, we can obtain one indicator to know when the project becomes stable. Specifically, we focus on OSS development using a bug tracking system in the operational phase, as shown in Figure 1. In this paper, we aim to predict the transition of the fix time using the exponential model and the delayed S-shaped model [13] derived from the software reliability growth models [14–17]. In particular, we predict the transition of the fix time considering the characteristics of open source projects. Then, we assume that the number of developers and users changes irregularly. We can easily discover the irregularity from various factors in open source projects and apply mathematical models with multiple parameters. However, it is difficult to use these models actually in terms of parameter estimation. In this paper, we apply a stochastic differential equation model with noise based on the Wiener process considering the specific circumstances of open source projects. The proposed model will be able to evaluate the project quantitatively considering external factors indirectly in open source projects.

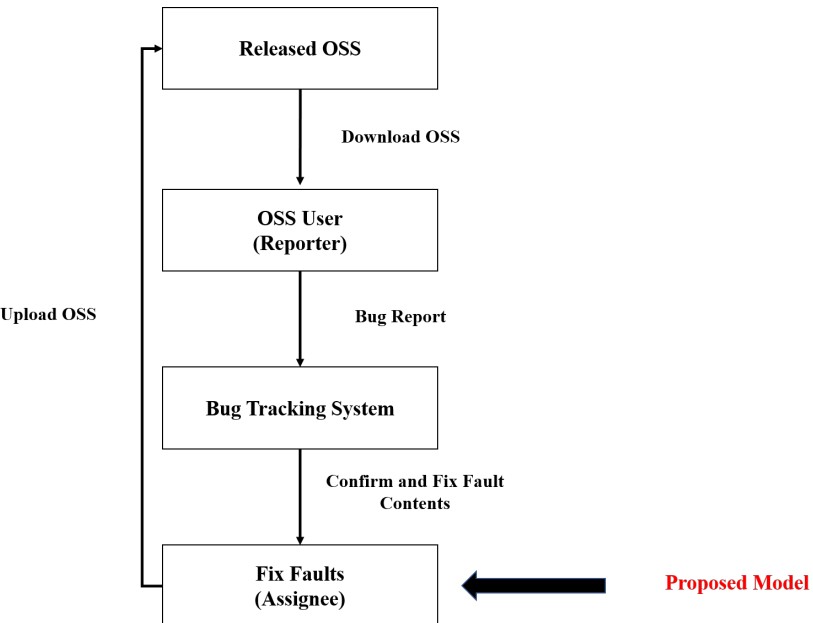

**Figure 1.** Open source software (OSS) development using the bug-tracking system.

Considering the characteristic of the method of fault fixing in open source projects, the time-dependent fix effort expenditure phenomenon maintains an irregular state because there is variability among the levels of developers and skill through version upgrades, and the OSS is developed and maintained by several developers and users.

## 3. Wiener Process Models for Estimating Fault Fix Time Transition

In Section 2, we discuss the stochastic differential equation modeling for the open source project. Thus, the operation phases of many OSS projects are influenced by external factors from triggers such as the difference of the skill of the fault assignee, the time lags of development, and maintenance activities. Based on the above points, we discuss the stochastic differential equation modeling for the OSS project. Let $\Omega(t)$ be the cumulative fault fixing time needed up to operational time $t$ ($t \geq 0$) from the start of the open source project. Suppose that $\Omega(t)$ takes on continuous real values. Since the estimated fault fix time is observed during the operational phase of the open source project, $\Omega(t)$ gradually increases as the operational procedures go on. Based on the software reliability growth modeling approach [14–17], the following linear differential equation in terms of fault fix time can be formulated:

$$\frac{d\Omega(t)}{dt} = \beta(t)\{\alpha - \Omega(t)\}, \tag{1}$$

where $\beta(t)$ is the increase rate of the fault fix time at operational time $t$ and a non-negative function and $\alpha$ means the estimated fault fix time required until the end of the operation.

Therefore, we extend Equation (1) to the following stochastic differential equation with Brownian motion [18]:

$$\frac{d\Omega(t)}{dt} = \{\beta(t) + \sigma\nu(t)\}\{\alpha - \Omega(t)\}, \tag{2}$$

where $\sigma$ is a positive constant representing the magnitude of the irregular fluctuation and $\nu(t)$ a standardized Gaussian white noise. By using Itô's formula [19], we can obtain the solution of Equation (2) under the initial condition $\Omega(0) = 0$ as follows:

$$\Omega(t) = \alpha[1 - \exp\{-\int_0^t \beta(s)ds - \sigma\omega(t)\}], \tag{3}$$

where $\omega(t)$ is a one-dimensional Wiener process, which is formally defined as an integration of the white noise $\nu(t)$ with respect to time $t$. Moreover, we define the increase rate of the fault fix time in the case of $\beta(t)$, defined as [20]:

$$\int_0^t \beta(s)ds \doteq \frac{\frac{dF_*(t)}{dt}}{\alpha - F_*(t)}. \tag{4}$$

In this paper, we assume the following equations based on software reliability models $F_*(t)$ as the cumulative fault fix time function of the proposed model:

$$F_e(t) \equiv \alpha(1 - e^{-\beta t}), \tag{5}$$

$$F_s(t) \equiv \alpha\left\{1 - (1 + \beta t)e^{-\beta t}\right\}, \tag{6}$$

where $\Omega_e(t)$ means the cumulative fault fix time for the exponential software reliability growth model with $F_e(t)$. Similarly, $\Omega_s(t)$ is the cumulative fault fix time for the delayed S-shaped software reliability growth model with $F_s(t)$.

Therefore, the cumulative fault fix time up to time $t$ is obtained as follows:

$$\Omega_e(t) = \alpha[1 - \exp\{-\beta t - \sigma \omega(t)\}], \tag{7}$$

$$\Omega_s(t) = \alpha[1 - (1 + \beta t)\exp\{-\beta t - \sigma \omega(t)\}]. \tag{8}$$

In this model, we assume that the parameter $\sigma$ depends on several noises due to external factors from several triggers in open source projects. Then, the expected cumulative fault fix time spent up to time $t$ is obtained as follows:

$$E[\Omega_e(t)] = \alpha[1 - \exp\{-\beta t + \frac{\sigma^2}{2}t\}], \tag{9}$$

$$E[\Omega_s(t)] = \alpha[1 - (1 + \beta t)\exp\{-\beta t + \frac{\sigma^2}{2}t\}]. \tag{10}$$

Similarly, we consider the sample path of the fault fixing time required for OSS maintenance, e.g., the needed remaining fault fixing time from time $t$ to the end of the project is obtained as follows:

$$\Omega_{re}(t) = \alpha \exp\{-\beta t - \sigma \omega(t)\}, \tag{11}$$

$$\Omega_{rs}(t) = \alpha(1 + \beta t)\exp\{-\beta t - \sigma \omega(t)\}. \tag{12}$$

Then, the expected fault fix time required for OSS maintenance until the end of operation time $t$ is obtained as follows:

$$E[\Omega_{re}(t)] = \alpha \exp\{-\beta t + \frac{\sigma^2}{2}t\}, \tag{13}$$

$$E[\Omega_{rs}(t)] = \alpha(1 + \beta t)\exp\{-\beta t + \frac{\sigma^2}{2}t\}. \tag{14}$$

Furthermore, we derive the sample path of the fault fixing time transition, called the density function of the cumulative fault fix time, from Equations (2), (7), and (8) as follows:

$$\frac{d\Omega_{re}(t)}{dt} = \{\beta(t) + \sigma v(t)\}\{\alpha \exp\{-\beta t - \sigma \omega(t)\}\}, \tag{15}$$

$$\frac{d\Omega_{rs}(t)}{dt} = \{\frac{\beta^2 t}{1 + \beta t} + \sigma v(t)\}\{\alpha(1 + \beta t)\exp\{-\beta t - \sigma \omega(t)\}\}. \tag{16}$$

By using the equations derived above, we can evaluate the stability degree of the open source project by predicting the transition of the fault fixing time, the cumulated fault fixing time, and the remaining fault fixing time.

## 4. Application of the Proposed Method to Actual Data

We discuss the proposed method by using actual project data. We also assess the stability of the project. In this section, we used the datasets of several OSS. To apply the prediction model to actual project data, we used the data from Eclipse and OpenStack. We can obtain large-scale datasets from these projects. The Eclipse and OpenStack projects use Bugzilla as the open source bug tracking system. In particular, we used Eclipse Version 4.7 (Oxygen) and OpenStack Version 16 (Pike) data. From the data used in this research, the Eclipse project had 1760 fault fix data, and the OpenStack project had 2249 fault fix data. Specifically, we used only the fixed faults as the number of fault fixes.

Figures 2 and 3 show the transition of the weekly average fault fixing time and the number of fault reports, respectively. Figures 2 and 3 show that the fault fix time decreased as the project progressed. Furthermore, the number of reported faults tended to decrease after the peak number of reports. Therefore, these projects are likely to become stable with the passage of time.

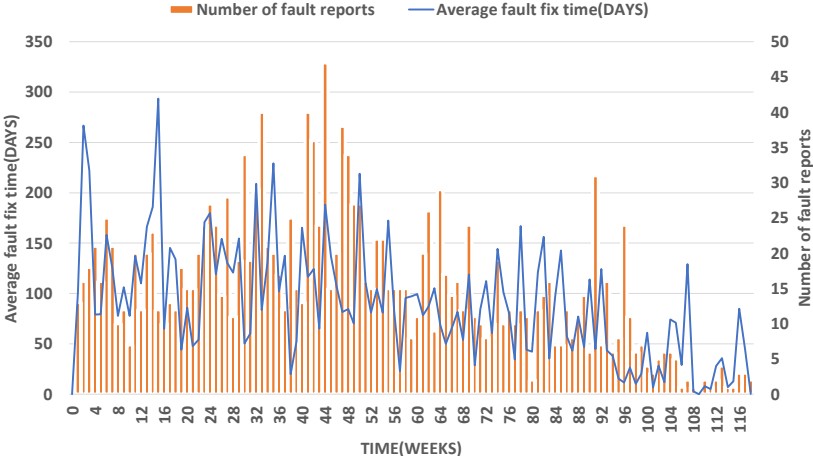

**Figure 2.** The transition of the weekly average fault fixing time and the number of fault reports for Eclipse.

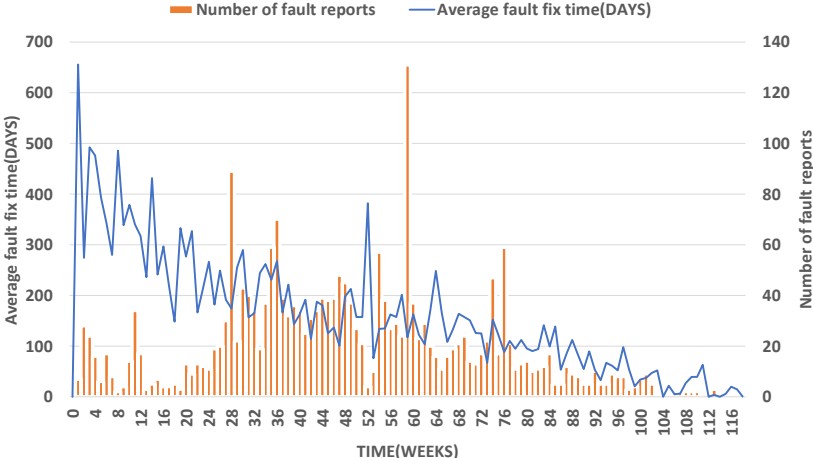

**Figure 3.** The transition of the weekly average fault fixing time and the number of fault reports for OpenStack.

In Figures 4–7, the "Sample path" is the required remaining fault fixing time at time $t$. The sample path depends on various factors in open source projects. The "Estimate" shows the expected value of the sample path. Furthermore, in these figures, data up to the dotted blue line were trained. In Figures 4–7, we can understand that both Eclipse and OpenStack data fit better models for the exponential models than the delayed S-shaped models. Furthermore, we can see from Table 1 that the exponential model fit better than the delayed S-shape in terms of AIC (Akaike's Information Criterion).

In addition, the exponential models estimated that the fix required more time than for the delayed S-shaped model. The sample path for each model had large noise at the start of the project because the estimate $\hat{\sigma}$ affected the size of the noise. In other words, we could estimate that these projects were unstable.

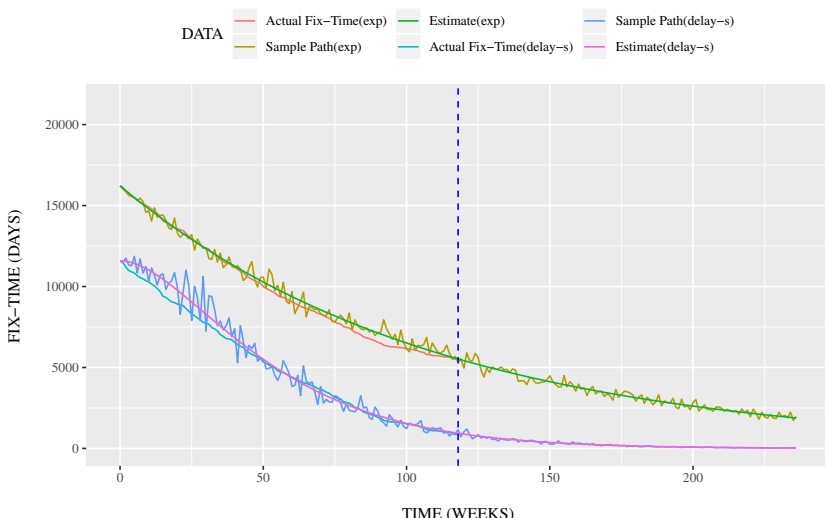

**Figure 4.** Remaining fault fixing time using the exponential model and the S-shaped model in Eclipse.

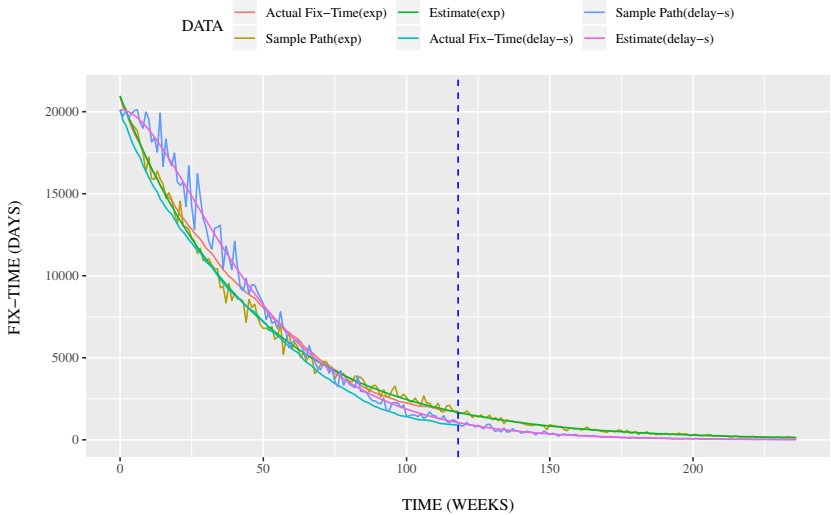

**Figure 5.** Remaining fault fixing time using the exponential model and the S-shaped model in OpenStack.

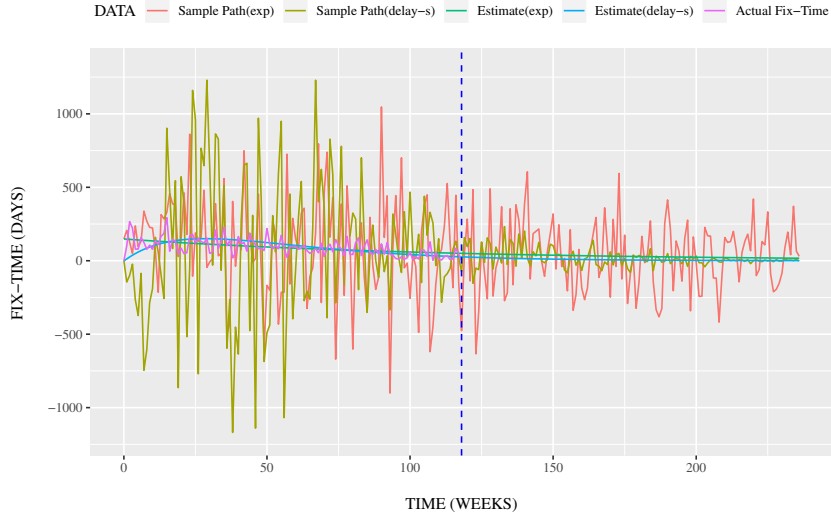

**Figure 6.** The transition of the fault fixing time using the exponential model and the S-shaped model in Eclipse.

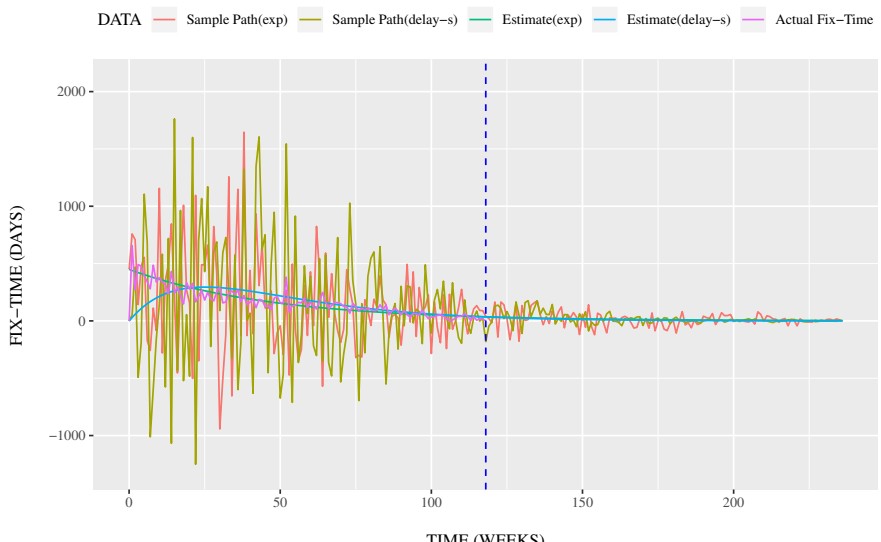

**Figure 7.** The transition of the fault fixing time using the exponential model and the S-shaped model in OpenStack.

**Table 1.** The parameter estimation in Eclipse and OpenStack data.

| | | Eclipse | | OpenStack | |
|---|---|---|---|---|---|
| | | **Exponential** | **Delayed S-Shaped** | **Exponential** | **Delayed S-Shaped** |
| parameter | $\alpha$ | $1.621 \times 10^4$ | $1.158 \times 10^4$ | $2.096 \times 10^4$ | $2.012 \times 10^4$ |
| | $\beta$ | $9.152 \times 10^{-3}$ | $3.535 \times 10^{-2}$ | $2.147 \times 10^{-3}$ | $3.987 \times 10^{-2}$ |
| | $\sigma$ | $5.248 \times 10^{-3}$ | $1.603 \times 10^{-2}$ | $8.431 \times 10^{-3}$ | $1.192 \times 10^{-2}$ |
| AIC | | 1274.079 | 1337.218 | 1283.291 | 1314.55 |

## 5. The Number of Data Required for Forecasting

It is necessary to make an accurate prediction as early as possible, if the project managers apply the proposed method to an actual project. Therefore, the data of 118 weeks used in this paper were divided into four phases. Then, the accuracy of the model was evaluated by changing the amount of training data. In the data of 118 weeks, the data not used as learning data were used as test data. Furthermore, the evaluation criterion was RMSE (Root Mean Squared Error) in the test data. The schematic of the evaluation method is shown in Figure 8. As shown in Figure 8, we gradually increased the amount of learning data. Then, the testing data would gradually decrease with the increasing amount learning data.

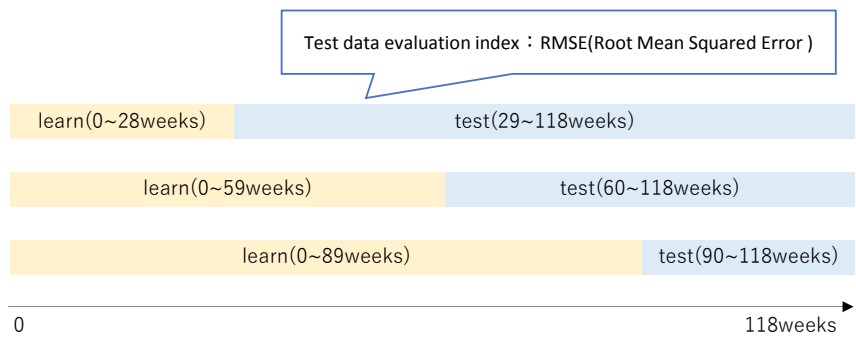

**Figure 8.** Evaluation method of prediction accuracy for the test data.

We applied the remaining fault fixing time and fault fixing time transition derived in Section 3 to Eclipse and OpenStack data. In this paper, we estimated the parameters by the method of the maximum likelihood. Furthermore, the unit of the fault fixing time was "days". Figures 4–7 show the estimation results of the remaining fault fixing time and the transition of the fault fixing time in cases of transition by using the Eclipse and OpenStack data.

Tables 2 and 3 show the results of model accuracy evaluation by using RMSE when changing the amount of training data. In addition, the results with the best value (minimum value) for RMSE are marked in bold. Both Tables 2 and 3 show that the exponential model fit better to the actual data in terms of RMSE. In particular, the RMSE of the transition in terms of faults remaining and fixing time in the case of the exponential model became small in the data of 58 weeks. Overall, we found that the exponential model was better than the delayed S-shaped model in terms of the prediction of the fault fixing time.

**Table 2.** Prediction accuracy by data volume change in Eclipse.

|  |  | Exponential | | | Delayed S-Shaped | | |
| --- | --- | --- | --- | --- | --- | --- | --- |
|  |  | AIC | RMSE (Remaining) | RMSE (Density) | AIC | RMSE (Remain) | RMSE (Density) |
| | 0~28 | 332.6751 | 964.499 | 46.35675 | 337.3291 | 3295.636 | 73.56478 |
| learning data | 0~58 | 657.3179 | **158.261** | **38.70379** | 672.7812 | 1122.064 | **47.77958** |
| (week) | 0~88 | 961.2937 | 426.7352 | 42.58499 | 1002.78 | **158.2334** | 55.55566 |
| | 0~118 | 1274.079 | - | - | 1337.218 | - | - |

**Table 3.** Prediction accuracy by data volume change in OpenStack.

|  |  | Exponential | | | Delayed S-Shaped | | |
| --- | --- | --- | --- | --- | --- | --- | --- |
|  |  | AIC | RMSE (Remaining) | RMSE (Density) | AIC | RMSE (Remain) | RMSE (Density) |
| | 0~28 | 343.7428 | 2322.344 | 58.17622 | 358.4923 | 6001.608 | 111.3953 |
| learning data | 0~58 | 686.2515 | **241.5871** | **33.97518** | 731.2697 | 1561.001 | 48.37366 |
| (week) | 0~88 | 977.9131 | 526.2862 | 37.40804 | 1032.3 | **182.6054** | **25.18201** |
| | 0~118 | 1283.291 | - | - | 1314.55 | - | - |

Next, Figures 9–12 show the results of predicting the remaining fault fix time and the fault fix time transition for Eclipse and OpenStack. Furthermore, in these figures, data up to the dotted blue line were trained. Although there was the problem that the noise was over-represented in the prediction of the fault fix time transition, the expected values of the remaining fault fix time and the fault fix time transition were both predicted accurately. Therefore, it was possible to predict the remaining fault fix time and the fault fix time transition with high accuracy with data of 58 weeks (about one year).

In addition, from Figures 13 and 14, it is possible to estimate roughly by learning 28 weeks of data (1/4 of all the data) in the fault fix time transition.

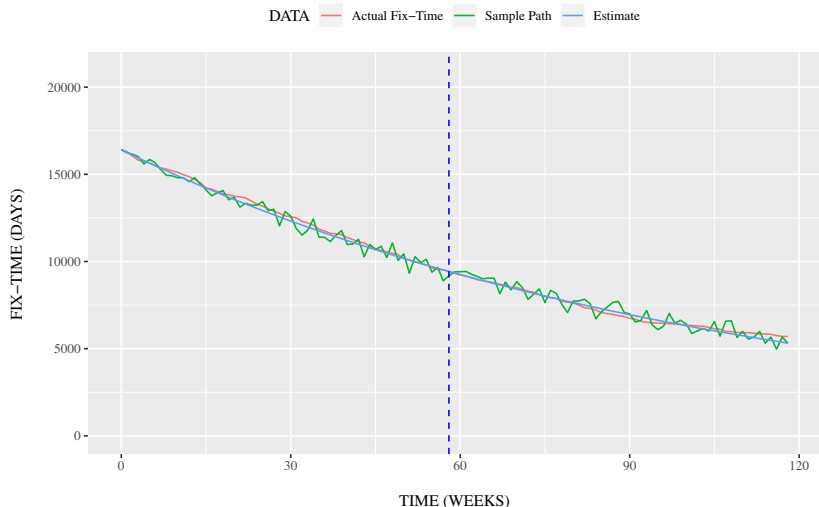

**Figure 9.** Remaining fault fixing time using the exponential model with 58 weeks o learning data in Eclipse.

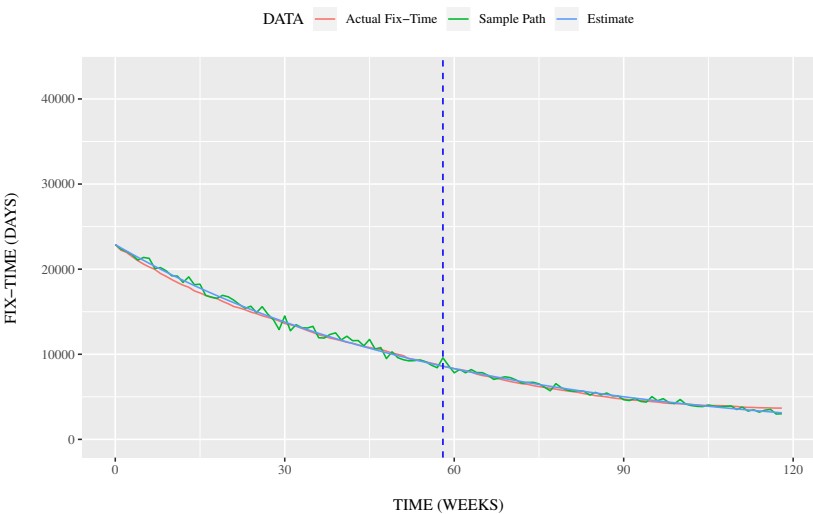

**Figure 10.** Remaining fault fixing time using the exponential model with 58 weeks of learning data in OpenStack.

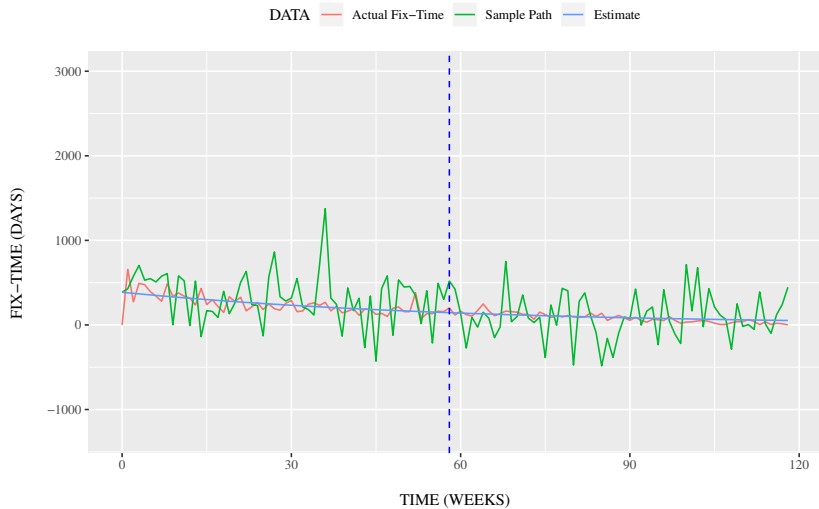

**Figure 11.** Fault fixing time transition using exponential model with 58 weeks learning data in Eclipse.

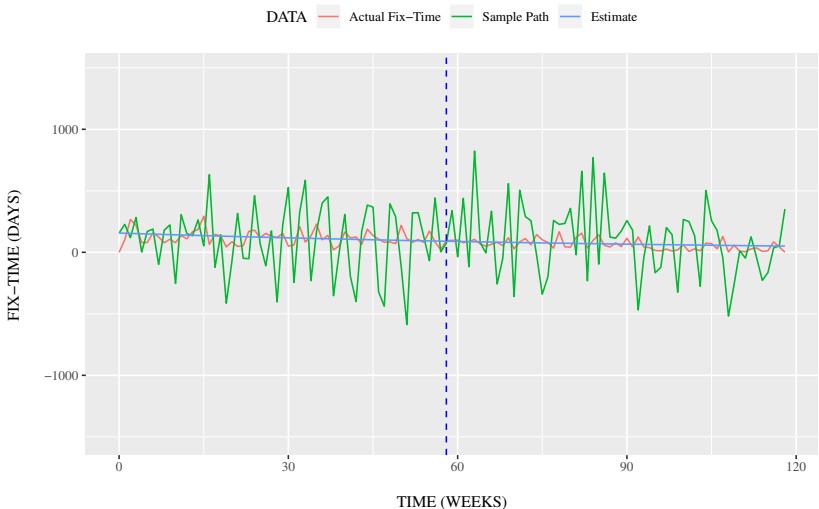

**Figure 12.** Fault fixing time transition using the exponential model with 58 weeks of learning data in Eclipse.

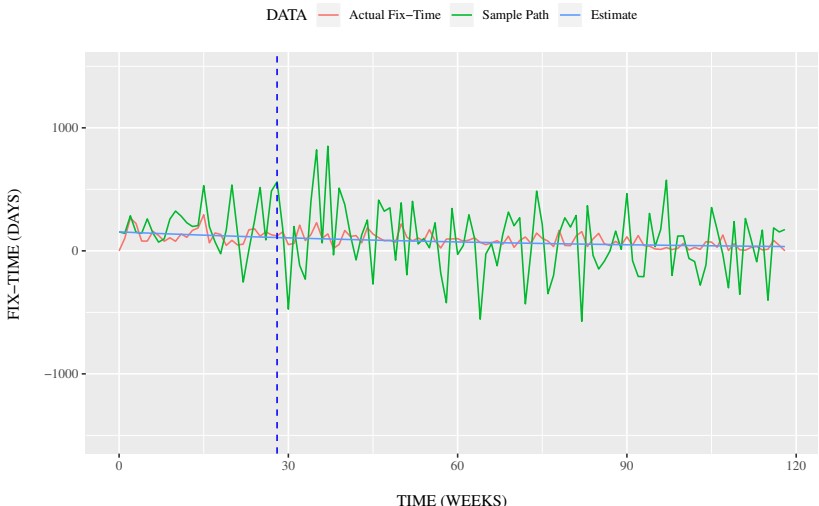

**Figure 13.** Fault fixing time transition using the exponential model with 28 weeks of learning data in Eclipse.

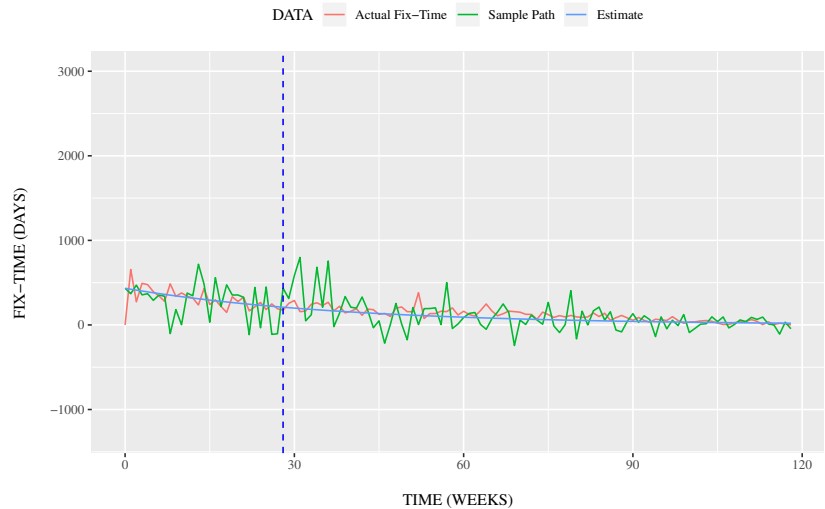

**Figure 14.** Fault fixing time transition using the exponential model with 28 weeks of learning data in Eclipse.

## 6. Conclusions

In general, the prediction of development effort and fixing time for individual faults can be assessed by using conventional OSS reliability evaluation methods. However, there is no research in terms of the transition of the fault fixing time for a long time, i.e., there is no research for predicting the fault fixing time. It is difficult to use the conventional software reliability growth model for the fault fix time, because the conventional software reliability growth models mainly evaluate the number of faults in software development. We can easily control open source projects if we can rapidly assess the stability and reliability of future projects by using fault fix time data. Thereby, the proposed method will lead to assessing the stability of OSS systems developed in terms of the convergence of the fault fixing time under various open source projects. Furthermore, the appropriate control of management effort for OSS will indirectly link to the quality, safety, reliability, and cost reduction of OSS if the manager knows the future of the project's progress.

In this paper, we discussed the transition analysis method of fault fixing time based on a stochastic differential equation model in an open source project. In particular, considering the characteristics of changes in the fault fixing time in large-scale open source projects and the complexity in OSS development, we predicted the transition of fault fixing time based on the Wiener process. Furthermore, we discussed the number of data needed to predict the fault fixing time. As a result, the OSS project managers can predict the future fault fixing time in a relatively short time. The proposed method can help project managers and OSS users as an evaluation method of open source project progress in the operation phase.

Considering the types of OSS, there are the derived OSS programs such as "StarOffice " originating from "LibreOffice" "StarOffice", "OpenOffice", and "Go–OpenOffice". In this case, it would be difficult to estimate the fix time using the proposed method for an OSS program including many reusable components, because the old reusable component has an impact on much of the fault fix time. In a future study, we will develop the estimation methods for an OSS program including several reusable components.

**Author Contributions:** Conceptualization, H.S.; Data curation, H.S.; Formal analysis, H.S. and Y.T.; Investigation, H.S.; Methodology, H.S. and Y.T.; Project administration, Y.T. and S.Y.; Software, H.S.; Supervision, Y.T. and S.Y.; Validation, Y.T. and S.Y.; Visualization, H.S.; Writing—original draft, H.S.; Writing—review & editing, Y.T. and S.Y.

**Funding:** This research received no external funding.

**Conflicts of Interest:** The authors declare no conflict of interest.

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
