# Peer review of "Prediction of Fault Fix Time Transition in Large-Scale Open Source Project Data"

_data, 2019_

Reviewer 1 Report

I only have a question, I don't know if I understood it well, but I am wondering if your predictor is able to estimate the number of faults found in an arbitrary OSS project at a specific time. Thus, e.g., we take the LibreOffice project, and we can predict the faults found in this project during this week with your predictor proposal. Is that right? If so, I would suggest you include an additional example which shows this in your paper, as it will clarify your work and enlighten its capabilities, added to the Eclipse and OpenStack examples you show to check it.

Author Response

Please see the attachment file "reviewer_1.pdf".

Reviewer 2 Report

GENERAL OVERVIEW

The authors of this paper predict the transition of fix time in large-scale open source projects. For prediction, the authors use the software reliability growth model based on the Wiener process considering that the fault fix time in open source projects changes depending on various factors such as the fault reporting time, the assignees for fixing faults.

WEAK POINTS:

The ideas presented in the paper are very interesting and the paper is well written. However, there are some open questions:

- Which are the main contributions of this paper? We suggest to include them in the introduction.

- It would be interesting to have the structure of the paper described at the end of the Introduction section.

- No future work is proposed in conclusions section.

In conclusion, the paper can be improved provided that the authors answer the above-mentioned questions and modify the paper according to the suggestions.

Author Response

Please see the attachment file "reviewer_2.pdf".

Reviewer 3 Report

The submitted paper is very interesting and it refers to the present problems. However, I would sugest some editorial changes. The chapter introduction should refer only for the general knowlegde in the analysed area. The methodology of the authors' investigation should be moved to the next chapters (see Figure 1). In next chapter - Chapter 3 the proposed model should be presented.

Chapter 3.2 has almost the same caption as Chapter 3 - please reformulate it.

I suggest to add one more keyword: open source software.

It is needed to add detail description to Figure 8.

The paper requires movement of Figures 12-14 before conclusions.

Why Wiener Process Models are choosen? Please comment it. Now this information is hiden in Chapter 2.

Did you compare the results of the investigation with other methods? It would be interesting to add it.

Are there any limitations in the application?

What do you mean by adding this info: "Sample Availability: Samples of the compounds ...... are available from the authors."

Please correct English in the paper:

"We discuss the applicability of the proposed method by using the proposed method and actual
91 project data. We also discuss the characteristics of data needed to evaluate the stability degree of the
92 project.
93 3.1. Used Data Sets
94 In this paper, we used"

When you write in past tenses please keep it in the whole paper.

Author Response

Please see the attachment file "reviewer_3.pdf".
